# A temporally restricted function of the dopamine receptor Dop1R2 during memory formation

Jenifer C Kaldun[†], Emanuele Calia[†], Ganesh Chinmai Bangalore Mukunda[†], Cornelia Fritsch, Nikita Komarov, Simon G Sprecher*

Department of Biology, University of Fribourg, Fribourg, Switzerland

## eLife Assessment

The authors design and implement an elegant strategy to delete genomic sequences encoding the dopamine receptor dop1R2 from specific subsets of mushroom body neurons (ab, a'b' and gamma) and show that while none of these manipulations affect short term appetitive or aversive memory, loss of dop1R2 from ab or a'b' block the ability of flies to display measurable forms of longer forms of memory. These findings are **important** in confirming and extending prior observations, and well supported by **convincing** evidence that build on precise techniques for genetic perturbation.

*For correspondence:
simon.sprecher@unifr.ch

[†]These authors contributed equally to this work

Competing interest: The authors declare that no competing interests exist.

**Abstract** Dopamine is a crucial neuromodulator involved in many brain processes, including learning and the formation of memories. Dopamine acts through multiple receptors and controls an intricate signaling network to regulate different tasks. While the diverse functions of dopamine are intensely studied, the interplay and role of the distinct dopamine receptors to regulate different processes is less well understood. An interesting candidate is the dopamine receptor Dop1R2 (also known as Damb), as it could connect to different downstream pathways. Dop1R2 is reported to be involved in forgetting and memory maintenance; however, the circuits requiring the receptors are unknown. To study Dop1R2 and its role in specific spatial and temporal contexts, we generated a conditional knockout line using the CRISPR-Cas9 technique. Two FRT sites were inserted, allowing flippase-mediated excision of the dopamine receptor in neurons of interest. To study the function of Dop1R2, we knocked it out conditionally in the mushroom body of *Drosophila melanogaster*, a well-studied brain region for memory formation. We show that Dop1R2 is required for later memory forms but not for short-term aversive or appetitive memories. Moreover, Dop1R2 is specifically required in the α/β-lobe and the α'/β'-lobe but not in the γ-lobe of the mushroom body. Our findings show a spatially and temporally restricted role of Dop1R2 in the process of memory formation, highlighting the differential requirement of receptors during distinct phases of learning.

## Introduction

The neuromodulator dopamine is involved in a plethora of brain functions. Among them are learning and memory, reinforcement signaling, reward, arousal, and motor functions (*Klein et al., 2019*; *Missale et al., 1998*; *Tritsch and Sabatini, 2012*). Perturbations in the dopaminergic system are associated with diseases like Parkinson's disease, addiction, depression, schizophrenia, and many more (*Klein et al., 2019*; *Missale et al., 1998*; *Tritsch and Sabatini, 2012*). Therefore, it is important to understand dopamine signaling in greater detail, allowing us to develop more effective therapeutics or preventive measures.

Dopamine is synthesized from tyrosine in dopaminergic neurons (DANs) and binds to G-protein-coupled receptors (GPCRs) (*Klein et al., 2019*; *Missale et al., 1998*; *Yamamoto and Seto, 2014*). These dopamine receptors have seven transmembrane domains, with an intracellular C-terminus and an extracellular N-terminus. The receptors interact with different G-proteins, which are heterotrimeric protein complexes consisting of an α-, β-, and γ-subunit. The G-protein complex binds to the third intracellular loop and the C-terminus of the receptors. Humans and other mammals express five different dopamine receptors, separated into two types. Type 1 dopamine receptors elevate protein kinase A (PKA) signaling and cyclic adenosine monophosphate (cAMP) levels via the α-subunit $G\alpha_s$, whereas type 2 receptors inhibit PKA activity via $G\alpha_{i/o}$, thus reducing cAMP levels (*Klein et al., 2019*; *Missale et al., 1998*; *Tritsch and Sabatini, 2012*). Downstream of cAMP and PKA, transcriptional activation mediated by cAMP response element-binding protein leads to the expression of immediate early genes as a response to dopamine signaling (*Carlezonjr et al., 2005*; *Neves et al., 2002*). This pathway is a key regulator of long-term memory (LTM) formation (*Alberini and Kandel, 2015*; *Kaldun and Sprecher, 2019*; *Kandel, 2012*). Moreover, dopamine receptors can modulate internal $Ca^{2+}$ levels by engaging the phospholipase C signaling pathway. Furthermore, the G-protein β- and γ-subunits can directly modulate voltage-gated and ligand-gated ion channels (*Klein et al., 2019*; *Ledonne and Mercuri, 2017*; *Missale et al., 1998*; *Nishi et al., 2011*; *Tritsch and Sabatini, 2012*). However, the coordination of the receptors and the different dopamine-mediated processes in specific circuits remains unexplored. Therefore, an accessible and well-characterized model is required, like the olfactory circuit of *Drosophila melanogaster*. The connectome of this circuit is known and can be manipulated with the sophisticated genetic tools of the fly (*Takemura et al., 2017*; *Li et al., 2020*; *Owald et al., 2015*). Similar to mammals, the fly uses dopamine in learning, memory, forgetting, negative and positive reinforcement, locomotion, and sleep and arousal regulation (*Berry et al., 2012*; *Burke et al., 2012*; *Karam et al., 2020*; *Sabandal et al., 2021*; *Sabandal et al., 2020*; *Siju et al., 2021*; *Siju et al., 2020*; *Sitaraman et al., 2015*; *Tomita et al., 2017*; *Waddell, 2013*; *Yamamoto and Seto, 2014*). Additionally, L-DOPA, the precursor for dopamine and a powerful receptor agonist, as well as other pharmaceutics, has been shown to function in the fly (*Yamamoto and Seto, 2014*). Thus, the dopaminergic system can be studied in *Drosophila*, taking advantage of the available genetic tools.

The fly uses four different dopamine receptors. Dop1R1 (Dumb) (*Kim et al., 2003*; *Sugamori et al., 1995*) and Dop1R2 (Damb) (*Feng et al., 1996*; *Han et al., 1996*) are type 1 receptors. Dop2R is a type 2 receptor (*Hearn et al., 2002*), and DopEcR is a type 1 dopamine receptor that also uses ecdysone as a ligand (*Srivastava et al., 2005*). All four receptors were shown to be involved in learning, memory, and forgetting (*Berry et al., 2012*; *Karam et al., 2020*; *Kim et al., 2007*; *Lark et al., 2017*; *Qi and Lee, 2014*; *Qin et al., 2012*; *Scholz-Kornehl and Schwärzel, 2016*; *Sun et al., 2020*; *Zhou et al., 2019*). While it is well established that Dop1R1 is crucial for learning and short-term memory (STM) (*Kim et al., 2007*; *Qin et al., 2012*), the role of the other receptors is less clear. Dop1R2 is an interesting candidate to study, due to its ability to couple to two different G-proteins, $G\alpha_s$, which engages the cAMP pathway, and $G\alpha_q$, which is involved in $Ca^{2+}$ signaling (*Han et al., 1996*; *Himmelreich et al., 2017*; *Sun et al., 2020*). This could allow Dop1R2 to modulate learning and memory in a complex fashion. The receptor is mainly expressed in the mushroom body (MB) (*Crocker et al., 2016*; *Croset et al., 2018*; *Han et al., 1996*; *Kim et al., 2007*; *Lark et al., 2017*), an important brain region for olfactory associative learning (*Aso et al., 2014a*; *Aso et al., 2014b*; *Cognigni et al., 2018*). Previous single-cell transcriptomics data show that Dop1R2 is sparsely found in the whole nervous system (in 7.99% of ventral nerve cord cells, and in 13.8% of cells outside the MB) (*Allen et al., 2020*; *Davie et al., 2018*). Work on a mutant line suggests that Dop1R2 is involved in forgetting (*Berry et al., 2012*). However, a study using RNAi silencing suggests that the receptor plays a role in memory maintenance (*Sun et al., 2020*). As these studies used different learning assays – aversive and appetitive, respectively, as well as different methods, it is unclear if Dop1R2 has different functions for the different reinforcement stimuli. To resolve this problem, we generated a transgenic line to conditionally knock out Dop1R2 in a spatially and temporally specific manner. Using CRISPR-Cas9 and homology-directed repair (HDR), we included FRT sites in the endogenous locus of Dop1R2 for flippase-mediated excision. In addition, we inserted an HA-tag to monitor the spatial localization of the receptor. This also allows us to visualize the efficiency of the flip-out. We used this line to study the role of the receptor in learning and memory in the MB for both aversive and appetitive conditioning. Upon flip-out of the receptor in the MB, 2 hr memory and LTM are impaired. Similar results are obtained when we flip out Dop1R2 specifically in the

α/β-lobes and α′/β′-lobes of the MB, which are involved in those memory phases. Therefore, Dop1R2 is required in the α/β-lobes and α′/β′-lobes for later memory forms.

## Results

### Generation of the Dop1R2 conditional knockout line

To be able to study both the requirement of Dop1R2 in specific neurons, as well as the localization of the receptor, we generated a transgenic line that allows defined spatial and temporal knockout of the receptor. In short, we inserted two FRT sites for flippase-mediated excision (*Golic and Lindquist, 1989*; *Gratz et al., 2014*), as well as a 3xHA-tag to study the localization of the receptor. The HA-tag was chosen to minimize interference with the receptor structure. A representation of the HA-tagged receptor is depicted in *Figure 1A*. The third intracellular loop between transmembrane domains (TMDs) five and six, as well as the C-terminal tail, is required for G-protein binding (*Missale et al., 1998*). *Figure 1B* gives an overview of the experimental strategy. The endogenous locus is cut twice by CRISPR-Cas9 to replace the first two coding exons of Dop1R2 with the corresponding sequence flanked by FRT sites and fused to the HA-tag, using a donor plasmid as template for HDR. The donor plasmid also contains the region directly upstream and downstream of the exchange site as homology arms to align the plasmid. The plasmid carrying the sequence for the two guide RNAs (gRNAs), as well as the donor plasmid, was injected into embryos with maternal Cas9 expression (nos>Cas9). Using the Gal4-UAS system to express flippase in neurons of interest, the receptor can be irreversibly flipped out in those neurons while keeping it functional in the rest of the brain. The first FRT site was placed in the intron before the first coding exon, which contains all seven TMDs (*Figure 1C*). Since Dop1R2 has three transcript isoforms with differences in the C-terminus, the second FRT site was placed at the end of the last common exon. Thus, FLP-mediated recombination will lead to the deletion of the two common coding exons, including all TMDs. Successful insertion was verified by sequencing.

### Dop1R2<sup>cko</sup> is expressed in the MB

To monitor the localization of the tagged dopamine receptors, we first stained brains with an antibody against the HA-tag in 1-week-old flies. In the y, w control line (*Figure 1D*), the HA-tag does not show any specific staining. In the Dop1R2$^{cko}$ line, we see a clear signal in the entire MB (*Figure 1E*). This matches previous reports, as well as single-cell RNAseq data, showing that Dop1R2 is expressed in the MB (*Croset et al., 2018*; *Han et al., 1996*).

Flipping the receptor out in the MB using the MB-specific OK107-Gal4 driver in combination with UAS-flp abolishes the signal, demonstrating the efficiency of the FRT sites (*Figure 1F*). Therefore, the flip-out system seems to work as planned.

### STM is not affected by the loss of Dop1R2

We wanted to see if flipping out Dop1R2 in the MB affects memory acquisition and STM by using classical olfactory conditioning. In short, a group of flies is presented with an odor coupled to an electric shock (aversive) or sugar (appetitive) followed by a second odor without stimulus. For assessing their memory, flies can freely choose between the odors either directly after training (STM) or at a later time point.

To ensure that the introduced genetic changes to the Dop1R2 locus do not interfere with behavior, we first checked the sensory responses of that line (*Figure 2—figure supplement 1A–D*). The Dop1R2$^{cko}$ line shows comparable odor responses (*Figure 2—figure supplement 1A and B*) as well as sugar and shock response (*Figure 2—figure supplement 1C and D*) to the control line. Aversive STM (*Figure 2—figure supplement 1E*), as well as aversive 2 hr memory (*Figure 2—figure supplement 1F*), is not significantly different from the y, w control line. Moreover, appetitive STM is also comparable to the control (*Figure 2—figure supplement 1G*). Thus, unsurprisingly, the introduced changes to Dop1R2 do not interfere with normal receptor function.

To test the requirement of Dop1R2 for STM, we assessed memory performance directly after training for flies with flipped-out Dop1R2 in the whole MB or in individual lobes, along with the parental controls. The flies were aged for a week before undergoing classical olfactory conditioning. Both aversive STM and appetitive STM were tested. But first, we ensured that the whole MB flip-out line responds normally to the used odors and stimuli. UAS-Flp/+;; Dop1R2$^{cko}$; OK107-Gal4/+ flies show

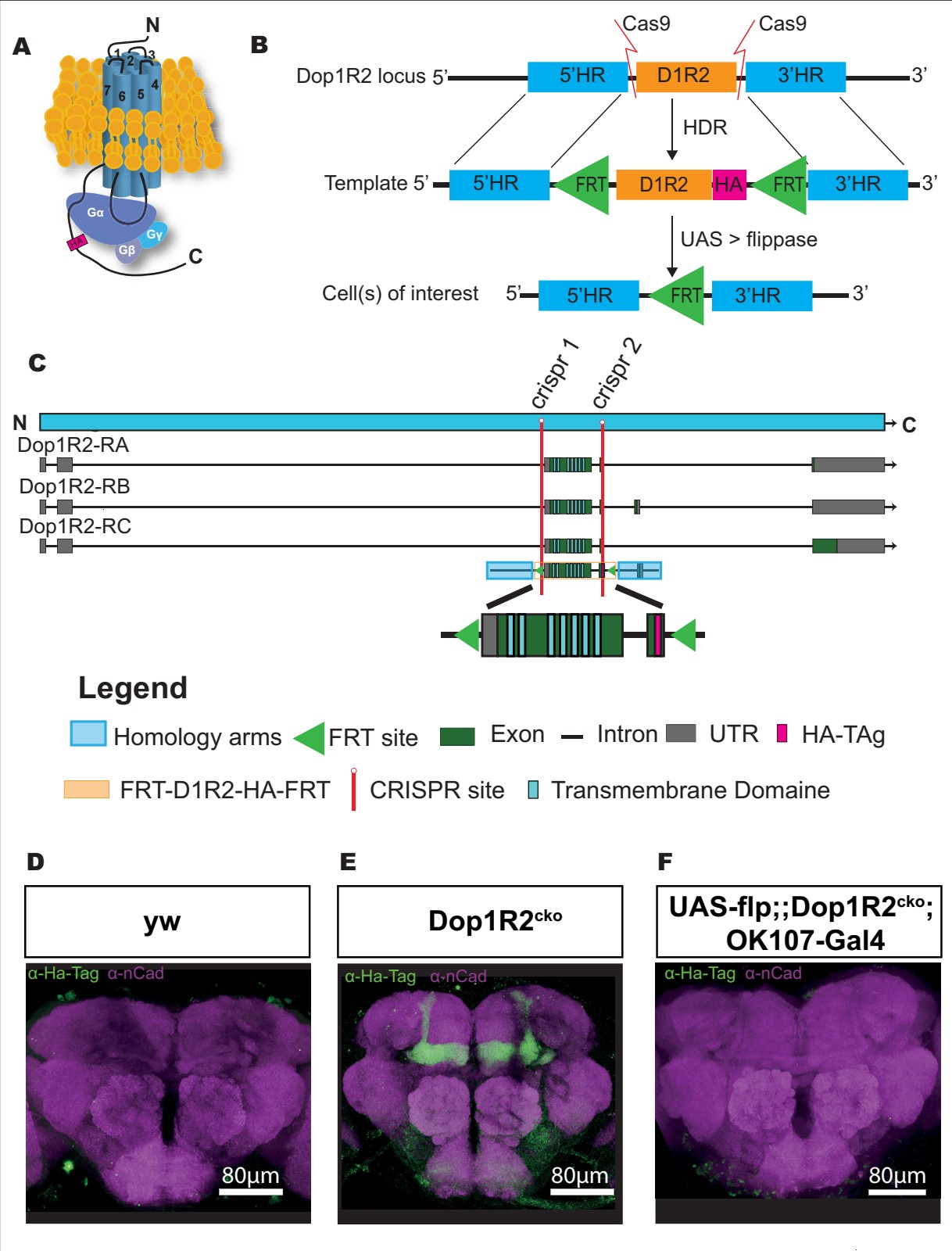

**Figure 1.** Generation of a conditional knockout allele for Dop1R2. (**A**) Schematic representation of the receptor structure and interaction with the G-protein complex. The position of the HA-tag is indicated. (**B**) Schematic representation of the conditional knockout system. The endogenous Dop1R2 was replaced using CRISPR-Cas9-mediated homology-directed repair (HDR) from a donor plasmid. The plasmid contained the two common coding exons of Dop1R2 with an HA-tag in the C-terminus and two FRT sites flanking this sequence. In the resulting Dop1R2^cko allele, the inserted HA-tag and

*Figure 1 continued on next page*

*Figure 1 continued*

Dop1R2 sequence can be removed by flippase (FLP) in cells of interest. (**C**) Schematic representation of the Dop1R2 gene locus with three different transcript isoforms. The location of the two used CRISPR sites is highlighted in red. The positions of the transmembrane domains in the isoforms and in the donor plasmid are indicated. (**D–F**) Dop1R2::HA expression in a frontal brain confocal section of (**D**) y, w, (**E**) Dop1R2^cko, or (**F**) *UAS-flp/+;; dop1R2^cko, OK107-Gal4/+* flies aged 1 week. The HA-tag was visualized using an anti-HA-tag antibody (green). Brain structures were labeled with anti-N-cadherin (nCad, magenta) antibody. Scale bar: 80 µm.

The online version of this article includes the following figure supplement(s) for figure 1:

**Figure supplement 1.** Single-cell transcriptomic analysis of Dop1R2 and various G-proteins in the mushroom body (MB) neurons.

comparable responses as the parental controls to both odors (*Figure 2—figure supplement 1H and I*), shock (*Figure 2—figure supplement 1J*) and sugar (*Figure 2—figure supplement 1K*).

For aversive conditioning, flipping out Dop1R2 in the whole MB does not change STM compared to parental controls (*Figure 2A*). Next, we flipped out Dop1R2 in the γ-lobe using 5HTR1B-Gal4. This MB lobe is involved in STM (*Blum et al., 2009*; *Trannoy et al., 2011*; *Heisenberg et al., 2000*). Matching the results for the whole MB flip-out, we do not see a change in the performance score (*Figure 2B*). Similar results are obtained when we flip out the receptor in the α/β-lobes using c739-Gal4 (*Figure 2C*) or α′/β′-lobes using c305a-Gal4 (*Figure 2D*).

Reward STM is also not changed when Dop1R2 is flipped out in the whole MB (*Figure 2E*), in the γ-lobe (*Figure 2F*), the α/β-lobes (*Figure 2G*), or the α′/β′-lobes (*Figure 2H*). Taken together, the results indicate that Dop1R2 is required for STM in none of the MB lobes.

## 2 hr memory is impaired by the loss of Dop1R2

As Dop1R2 was previously described to be involved in forgetting and/or memory maintenance, we wanted to assess later time points after training.

First, we looked at 2 hr after aversive training. Flip-out in the whole MB leads to a reduced performance score (*Figure 3A*). Next, we asked which MB lobe might cause this reduction. Therefore, we flipped out Dop1R2 in the γ-lobe using the 5HTR1B-Gal4 driver, characterized in *Aso et al., 2012*, in the α/β-lobes using c739-Gal4, or α′/β′-lobes using c305a-Gal4. Loss of Dop1R2 in the γ-lobe does not reduce memory performance (*Figure 3B*). However, loss of Dop1R2 in both the α/β-lobes (*Figure 3C*) or the α′/β′-lobes (*Figure 3D*) impaired 2 hr memory after aversive training.

We tested the same for reward memory using sugar as reinforcement. Flip-out of Dop1R2 in the whole MB (*Figure 3E*) results in a reduced performance score. Loss of Dop1R2 in the γ-lobe does not affect memory performance (*Figure 3F*). As for aversive training, flip-out of Dop1R2 in the α/β-lobes (*Figure 3G*) or the α′/β′-lobes (*Figure 3H*) impaired 2 hr memory after reward training. However, the reduction is not as severe as for aversive training. Taken together, the results indicate that Dop1R2 is required for aversive 2 hr memory as well as reward 2 hr memory in the α/β-lobes and the α′/β′-lobes but is dispensable in the γ-lobe.

## 24 hr memory is impaired by the loss of Dop1R2

Next, we wanted to see if later memory forms are also affected. One cycle of reward training is sufficient to create LTM (*Krashes and Waddell, 2008*), while for aversive memory, five to six cycles of electroshock training are required to obtain robust LTM scores (*Tully et al., 1994*). So, we looked at both 24 hr aversive and appetitive memory. For aversive LTM, the flies were tested on the Y-Maze apparatus as described in *Mohandasan et al., 2022*.

Flipping out Dop1R2 in the whole MB causes reduced 24 hr memory performance (*Figure 4A and E*). No phenotype was observed when Dop1R2 was flipped out in the γ-lobe (*Figure 4B and F*). However, similar to 2 hr memory, loss of Dop1R2 in the α/β-lobes (*Figure 4C and G*) or the α′/β′-lobes (*Figure 4D and H*) causes a reduction in memory performance. Thus, Dop1R2 seems to be involved in aversive and appetitive LTM in the α/β-lobes and the α′/β′-lobes.

Previous studies have shown that mutation in the Dop1R2 receptor leads to improvement in LTM when a single shock training paradigm is used (*Berry et al., 2012*). As we found that it disrupts LTM, we wanted to verify if the absence of Dop1R2 outside the MB is what leads to an improvement in memory. To that extent, we tested pan-neuronal flip-out of Dop1R2 flies for 6 hr and 24 hr memory upon single shock using the elav-Gal4 driver. We found that it did not improve memory at both time points (*Figure 4—figure supplement 1*), confirming that flipping out Dop1R2 pan-neuronally

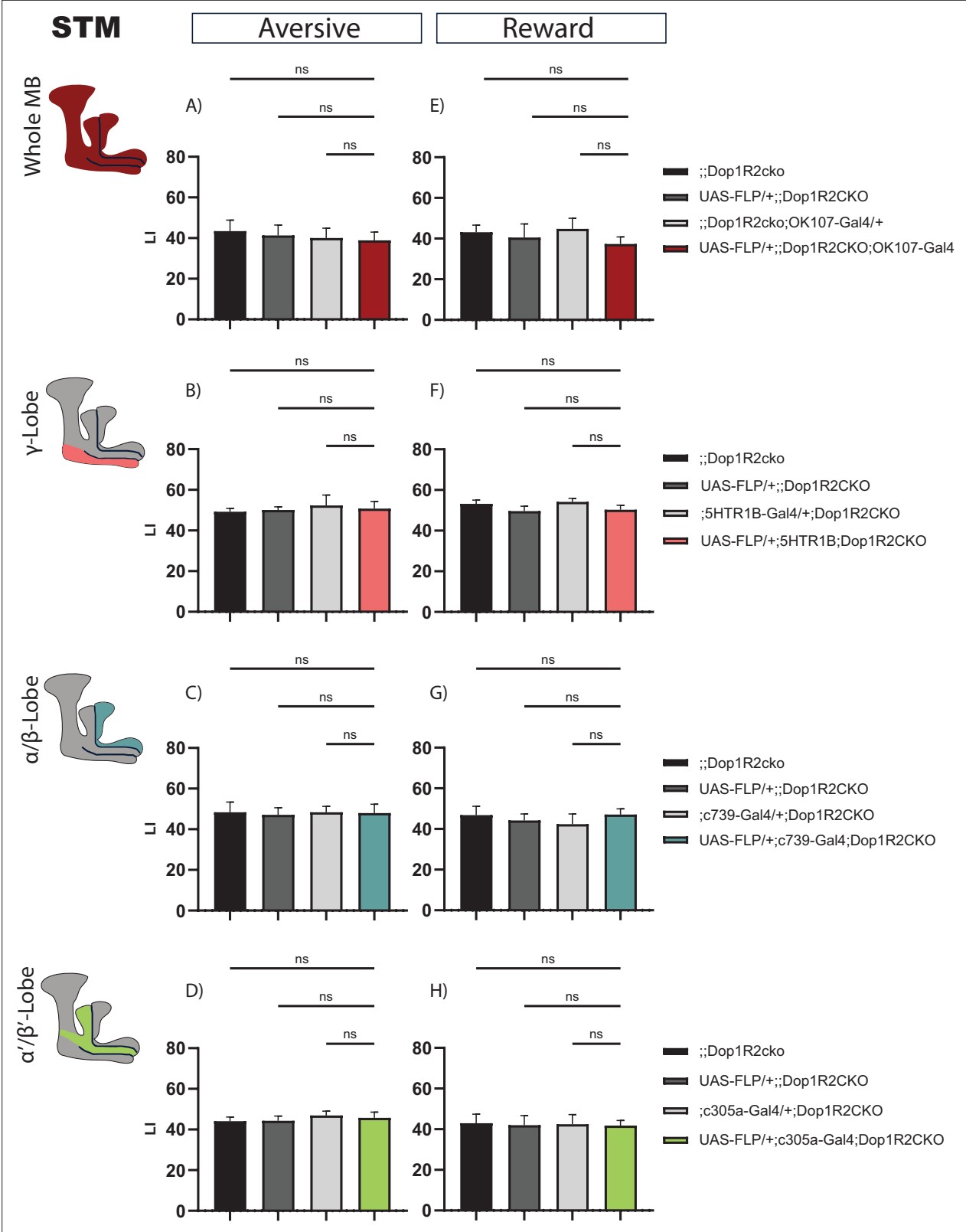

**Figure 2.** Short-term memory of flies with knockout of Dop1R2 in the mushroom body (MB). (**A–D**) Aversive training, (**E–H**) reward training. (**A** and **E**) Whole MB flip-out using OK107-Gal4 and parental controls. (**B** and **F**) γ-Lobe flip-out using 5HTR1B-Gal4 and parental controls. (**C** and **G**) α/β-Lobe flip-out using c739-Gal4 and parental controls. (**D** and **H**) α'/β'-Lobe flip-out using c305a-Gal4 and parental controls. No performance impairment was

*Figure 2 continued on next page*

*Figure 2 continued*

observed in any of the tested conditions. See **Figure 2—figure supplement 1** for sensory controls . Bar graphs represent the mean, and error bars represent the standard error of the mean. For each shown graph, N=12. ns: not significantly determined by a one-way ANOVA and Tukey's HSD.

The online version of this article includes the following figure supplement(s) for figure 2:

**Figure supplement 1.** Sensory tests of the Dop1R2 conditional knockout line.

does not improve LTM (*Figure 4—figure supplement 1C*) and highlighting its irrelevance in memory outside the MB.

## Developmental defects are ruled out in a temporally restricted Dop1R2 conditional knockout

To exclude developmental defects in the MB caused by flip-out of Dop1R2, we stained fly brains with a FasII antibody. Compared to genetic controls, the anatomical organization as judged by FasII staining in flies lacking Dop1R2 in the MB was not altered (*Figure 4—figure supplement 2C*).

To further provide behavioral evidence that the learning defects we observed are not due to developmental knockout of Dop1R2, we generated a Gal80$^{ts}$-containing line, enabling the temporal control of Dop1R2 knockout in the entire MB. Given that the half-life of the receptor remains unknown, we assessed both aversive STM and LTM to determine whether post-eclosion ablation of Dop1R2 in the MB produced differences compared to our previously tested line, in which Dop1R2 was constitutively knocked out from fertilization. To achieve this, flies were maintained at 18°C until eclosion and subsequently shifted to 30°C for 5–7 days. On the fifth day, training was conducted, followed by memory testing. Our results indicate that aversive STM was not significantly impaired in Dop1R2-deficient MBs compared to control flies (*Figure 4—figure supplement 3*), consistent with our previous findings (*Figure 2*). However, aversive LTM was significantly impaired relative to control lines (*Figure 4—figure supplement 3*), which also aligned with prior observations. These findings strongly indicate that memory loss caused by Dop1R2 flip-out is not due to developmental defects.

## Discussion

We have generated a conditional knockout line for the dopamine receptor Dop1R2 following a similar approach as in *Widmer et al., 2018*. To achieve this, FRT sites were inserted in front of the start codon and in the C-terminus of Dop1R2 using CRISPR-Cas9-mediated HDR. In addition, an HA-tag was inserted to be able to visualize the dopamine receptor expression, as well as verify successful flip out. Using an anti-HA-tag antibody, we were able to visualize Dop1R2 in the MB of the generated line. This matches previous reports for the expression of Dop1R2 (*Crocker et al., 2016*; *Croset et al., 2018*; *Han et al., 1996*; *Kim et al., 2007*; *Sun et al., 2020*). Moreover, upon knocking out Dop1R2 with an MB-specific driver, the HA-tag labeling disappears, indicating that the conditional knockout system works. The HA-tag could also be useful to study the subcellular localization of Dop1R2 within the MB lobes, e.g., if it is close to synapses of DANs.

To get a better overview of Dop1R2's role in the MB, we analyzed appetitive memory at different time points after training in the individual MB lobes. Loss of Dop1R2 in the whole MB, as well as the α/β-lobe or the α'/β'-lobe, impairs 24 hr reward memory. This observation matches previous studies. Using Dop1R2-RNAi in the MB, *Sun et al., 2020*, showed that STM is intact, while LTM is impaired. When knocking down the Raf/MAPK pathway, they get a similar phenotype. Moreover, expression of a constitutively active Raf allele rescues the Dop1R2-dependent LTM deficit, and Dop1R2 seems to be required for the phosphorylation of the MAPK. Therefore, Dop1R2-dependent activation of the Raf/MAPK pathway is required for stabilization of reward LTM memory. Another study shows that reward LTM requires food with a high energetic value (*Musso et al., 2015*). This signal seems to be relayed by the DAN MB-MP1 (PPL1-γ1pedc), which showed temporally restricted oscillating activity early post-training (*Musso et al., 2015*; *Pavlowsky et al., 2018*). Furthermore, knockdown of Dop1R2 using RNAi impaired reward LTM while leaving STM intact. Therefore, dopaminergic signaling through MB-MP-1 and Dop1R2 could indicate the energetic value of the reward and decide if LTM should be formed or not. Interestingly, the MB-MP1 DAN arborizes in the spur of the γ-lobe, as well as the inner core of the peduncle, which consists of the axons of the α/β Kenyon cells (KCs) (*Tanaka*

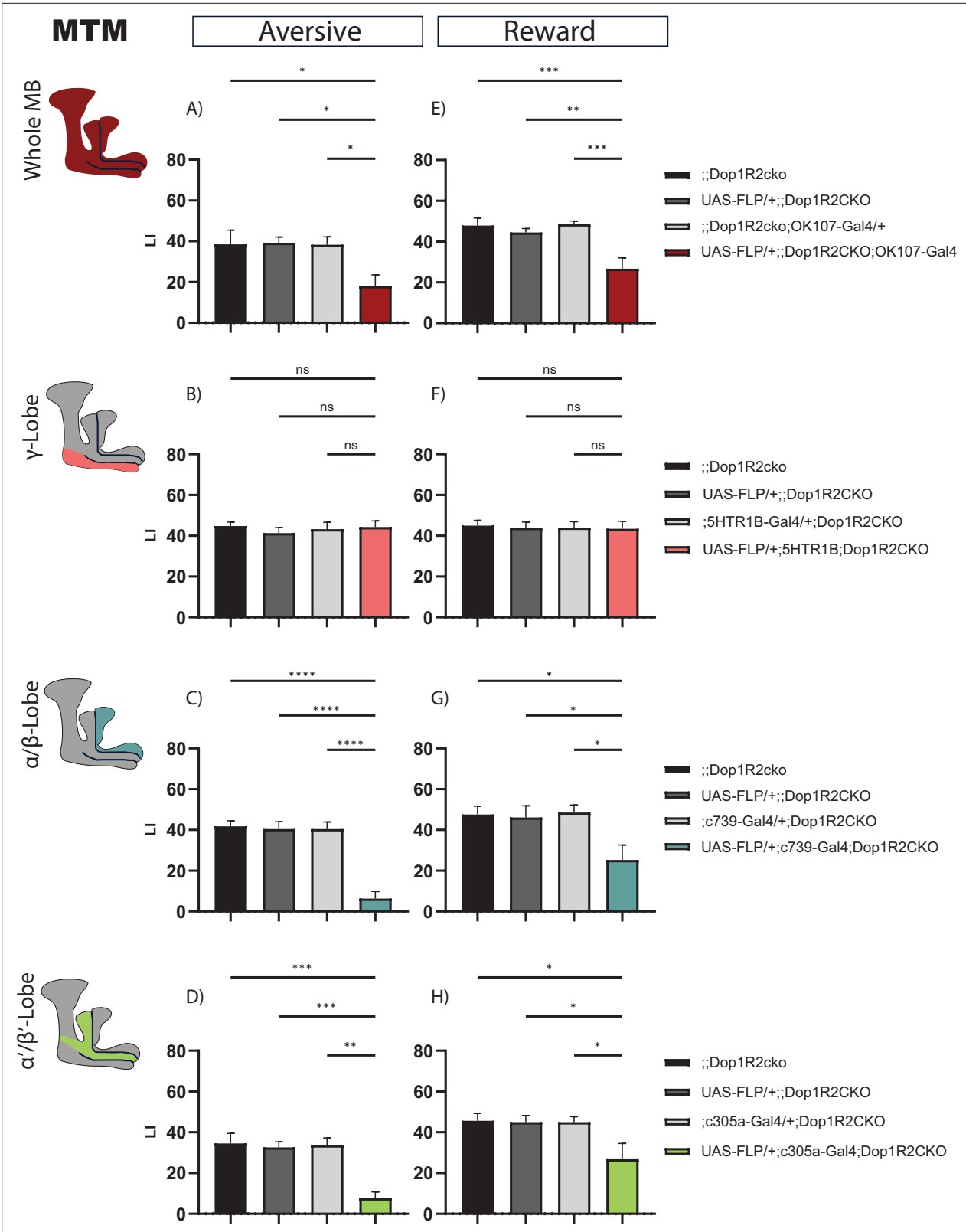

**Figure 3.** 2 hr memory of flies with knockout of Dop1R2 in the mushroom body (MB). (**A–D**) Aversive training, (**E–H**) reward training. (**A** and **E**) Whole MB flip-out using OK107-Gal4 and parental controls. (**B** and **F**) γ-Lobe flip-out using 5HTR1B-Gal4 and parental controls. (**C** and **G**) α/β-Lobe flip-out using c739-Gal4 and parental controls. (**D** and **H**) α′/β′-Lobe flip-out using c305a-Gal4 and parental controls. For whole MB flip-out, α/β-lobes and α′/β′-lobes, both aversive and appetitive 2 hr memory performance is impaired. Loss of Dop1R2 in the γ-lobe does not affect 2 hr memory. See *Figure 2—figure*

*Figure 3 continued on next page*

Figure 3 continued

**supplement 1** for sensory controls. Bar graphs represent the mean, and error bars represent the standard error of the mean. For each shown graph, N=12. Asterisks denote significant differences between groups (*p<0.05, **p<0.005, ***p<0.001, ****p<0.0001, ns: not significant) determined by one-way ANOVA and Tukey's HSD (panels **A–C**, **E–H**) and Kruskal-Wallis with Dunn's multiple comparisons test due to non-normal distribution (panel **D**).

*et al., 2008*), which according to our results require Dop1R2 for reward LTM. Loss of Dop1R2 in the MB output neuron MVP2 (MBON-γ1pedc>α/β) also impaired reward LTM while leaving STM intact (*Pavlowsky et al., 2018*). This GABAergic MVP2 MBON forms a feedback circuit with the MB-MP1 DAN. After training, the oscillatory activity of MB-MP1 is enhanced, while MVP2 is inhibited. After 30 min, MVP2 gets activated, and MB-MP1 is inhibited. Moreover, Dop1R2 seems to be required for modulating this feedback loop.

The MBONs are an important contributor to memory. Moreover, MBONs receive dopaminergic input and seem to express dopamine receptors (*Crocker et al., 2016*).

We showed that Dop1R2 is required in the α/β-lobe and the α'/β'-lobe for reward LTM formation. Interestingly, MBONs, which are involved in reward LTM (*Ichinose et al., 2015*; *Owald et al., 2015*; *Plaçais et al., 2013*), have arborization in the α/β-lobe and the α'/β'-lobe. Thus, Dop1R2 could modulate these connections as well.

In all, Dop1R2 is required for reward LTM formation, and loss of the receptor impairs LTM. Dop1R2 seems to influence reward LTM in different ways. First, by acting on the Raf-MAPK pathway to stabilize memory; second, by relaying the energetic content of the reward; and third, by modulating the MB-MP1-MVP2 loop.

For aversive conditioning, we observe that loss of Dop1R2 in the MB leads to impaired 2 hr memory and LTM, whereas STM is intact. Moreover, Dop1R2 seems to be required in the α/β-lobe and the α'/β'-lobe. A previous study using a mutant for Dop1R2, which also affects the neighboring gene GC1907, observed higher memory retention (*Berry et al., 2012*). Further, lack of Dop1R2 impairs reversal learning. It is proposed that Dop1R2 acts on the RAC-forgetting pathway (*Cervantes-Sandoval et al., 2016*; *Shuai et al., 2010*). Interestingly, Dop1R2 can act on two different downstream second messenger systems by using two different G-proteins (*Himmelreich et al., 2017*). By coupling to $G_{\alpha s}$, the cAMP pathway is activated. By coupling to $G_{\alpha q}$, the $Ca^{2+}$ messenger system is activated. Furthermore, knocking down $G_{\alpha q}$ pan-neuronally or in the MB leads to memory enhancement 3 hr after training, but not 6 hr after training (*Himmelreich et al., 2017*). Thus, Dop1R2 could regulate the RAC forgetting pathway through $G_{\alpha q}$.

Some RNA-binding proteins and immediate early genes help maintain identities of MB cells and are regulators of local transcription and translation (*de Queiroz et al., 2025*). So, the availability of different G-proteins may change in different lobes and during different phases of memory. The G-protein via which GPCRs signal may depend on the pool of available G-proteins in the cell/subcellular region (*Hermans, 2003*). Therefore, Dop1R2 may signal via different G-proteins in different compartments of the MB and also different compartments of the neuron. We looked at $G_{\alpha o}$ and $G_{\alpha q}$ as they are known to have roles in learning and forgetting (*Ferris et al., 2006*; *Himmelreich et al., 2017*). We found that Dop1R2 co-expresses more frequently with $G_{\alpha o}$ than with $G_{\alpha q}$ (*Figure 1—figure supplement 1*). While there is evidence for Dop1R2 to act via $G_{\alpha q}$ (*Himmelreich et al., 2017*), it is difficult to determine whether this interaction is exclusive or if Dop1R2 can also be coupled to other G-proteins. It will be interesting to determine the breadth of G-proteins that are involved in Dop1R2 signaling.

Berry and colleagues also observed that blocking the output of DANs inhibits forgetting, while activating DANs accelerates forgetting (*Berry et al., 2012*). This modulation seems to require ongoing activity of the MP1 DAN together with further DANs. Interestingly, using a spaced training protocol and a Dop1R2 RNAi knockdown, another study showed impaired LTM (*Plaçais et al., 2017*). Furthermore, after spaced training, flies have a higher energy uptake, and the energy metabolism is upregulated in the MB. This increase in energy consumption is mediated by dopaminergic signaling from the MB-MP1 DAN. Loss of Dop1R2 in the MB abolishes the increase in energy consumption. Therefore, Dop1R2 seems to be important for aversive LTM formation by regulating energy consumption. Thus, like for reward LTM formation, aversive LTM seems to require sufficient energy. Starved flies reduce the formation of aversive LTM (*Plaçais and Preat, 2013*; *Plaçais et al., 2012*). This information seems to be relayed through ongoing oscillation of MB-MP1 and Dop1R2

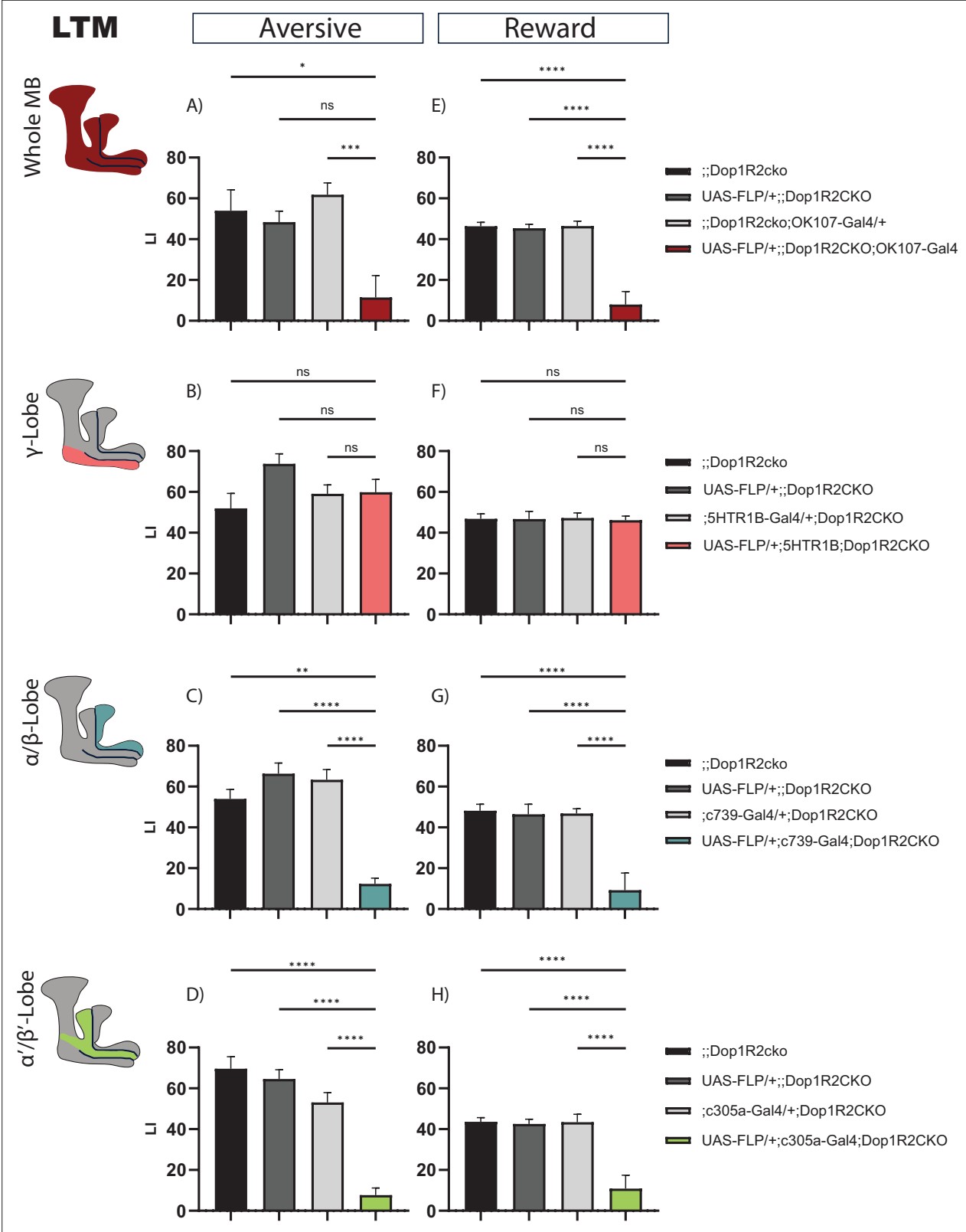

**Figure 4.** 24 hr memory of flies with knockout of Dop1R2 in the mushroom body (MB). (**A–D**) Aversive training, (**E–H**) reward training. (**A** and **E**) Whole MB flip-out using OK107-Gal4 and parental controls. (**B** and **F**) γ-Lobe flip-out using 5HTR1B-Gal4 and parental controls. (**C** and **G**) α/β-Lobe flip-out using c739-Gal4 and parental controls. (**D** and **H**) α'/β'-Lobe flip-out using c305a-Gal4 and parental controls. For whole MB flip-out, α/β-lobes and α'/β'-lobes, both aversive and appetitive 24 hr memory performance is impaired. Loss of Dop1R2 in the γ-lobe does not affect 24 hr memory. See *Figure 2—*

*Figure 4 continued on next page*

*Figure 4 continued*

*figure supplement 1* for sensory controls. Bar graphs represent the mean, and error bars represent the standard error of the mean. For each shown graph included in the reward training experiment, N=12, while for the graphs included in the aversive training experiment, N=14. Asterisks denote significant differences between groups (*p<0.05, **p<0.005, ***p<0.001, ****p<0.0001, ns: not significant) determined by Kruskal-Wallis with Dunn's multiple comparisons test due to non-normal distribution (panels **A–C**) and one-way ANOVA and Tukey's HSD (panels **D–H**).

The online version of this article includes the following figure supplement(s) for figure 4:

**Figure supplement 1.** Memory retention upon single shock aversive training in Dop1R2 pan-neuronal knockout flies.

**Figure supplement 2.** γ-Lobe development in the Dop1R2$^{cko}$ carrying flies.

**Figure supplement 3.** A temporally restricted knockout of Dop1R2 in the mushroom body rules out developmental defects.

after training. In addition, the MVP2 MBON might also be involved (*Ueoka et al., 2017*). Therefore, the gating mechanism in both aversive and reward LTM formation seems to require Dop1R2 (*Pavlowsky et al., 2018*).

In addition, the MAPK signaling pathway might also be required in aversive LTM formation by activating transcription factors like CREB and c-fos (*Miyashita et al., 2018*).

Both Dop1R2 and the ongoing activity of MB-MP1 seem to have multiple roles directly after training in a short time window (*Berry et al., 2012*; *Plaçais et al., 2017*; *Plaçais et al., 2012*). The circuit acts like a gating mechanism for LTM to ensure that there is enough energy for continuing LTM formation. The MBON MVP2 acts as a feedback loop to regulate the activity of the DAN. Moreover, both the RAC and the Raf/MAPK signaling pathway seem to be engaged to either forget or stabilize the memory. In aversive memory formation, loss of Dop1R2 could lead to enhanced or impaired memory, depending on the activated signaling pathways. The signaling pathway that is activated further depends on the available pool of secondary messengers in the cell (*Hermans, 2003*), which may be regulated by the internal state of the animal.

However, it remains unclear how all of these aspects are integrated and if there is a hierarchical order.

DANs can produce aversive and appetitive associations depending on the temporal presentation of odor cue and reinforcement stimulus (*Handler et al., 2019*). Thus, dopaminergic signaling can modify the KC-MBON synapses bidirectionally. The Dop1R1-G$_{αs}$-cAMP pathway seems to detect the temporal coincidence of the stimuli, whereas the Dop1R2-G$_{αq}$-Ca$^{2+}$ pathway detects the temporal ordering (*Handler et al., 2019*). Dop1R2 mutant flies seem to be able to form an odor association but are not able to update it. Further, Dop1R2 is required for the potentiation of the KC-MBON synapse following backward pairing.

Taken together, Dop1R2 has multiple roles during memory formation and integrates different signals, including the detection of the order of stimuli, the internal state, like energy levels and forgetting and maintenance signals.

The receptor does not seem to be required for STM but for later time points. Previous studies looking at the temporal requirement of the lobes (*Guven-Ozkan and Davis, 2014*; *Perisse et al., 2013*) defined the γ-lobe to be responsible for memory acquisition (*Blum et al., 2009*; *Trannoy et al., 2011*; *Heisenberg et al., 2000*). The α/β-lobe and its output are involved in LTM (*Akalal et al., 2011*; *Blum et al., 2009*; *Cervantes-Sandoval et al., 2013*; *Huang et al., 2012*; *Ichinose et al., 2015*; *Krashes and Waddell, 2008*; *Trannoy et al., 2011*). The function of the α'/β'-lobe seems to be LTM consolidation (*Cervantes-Sandoval et al., 2013*; *Krashes and Waddell, 2008*). However, both lobes also seem to have a role in middle-term memory (*Bouzaiane et al., 2015*; *Scheunemann et al., 2012*; *Shyu et al., 2019*; *Turrel et al., 2022*). As the loss of Dop1R2 in the γ-lobe or the whole MB does not impair STM, we conclude that the receptor is not required for this memory type. However, Dop1R2 is expressed in the γ-lobe (*Crocker et al., 2016*; *Croset et al., 2018*), so it might be required for other types of behaviors.

The impairment of reward LTM upon knockout of Dop1R2 in the α/β-lobe and the α'/β'-lobe matches the described role of these neurons. So, both the α/β-lobe and the α'/β'-lobe require Dop1R2 for LTM. Interestingly, both lobes also seem to require Dop1R2 for 2 hr memories. As the MB-MP1 DAN is active in this time window as well, this would suggest that Dop1R2 function at this time point is important for correct LTM formation.

This would indicate that Dop1R1 is the main contributor to STM while Dop1R2 is responsible for later memory stages. It would be interesting to know how the switch from Dop1R1 dependency to Dop1R2 occurs.

As Dop1R2 is required in the MBON MVP2 for reward LTM, it would be exciting to see if it is also required in other MBONs or neurons outside the MB. Besides learning and memory, the MB also uses dopamine to regulate sleep. This tool offers the opportunity to study both aspects in neurons of interest.

The genetic tool we generated here to study the role of the Dop1R2 dopamine receptor in cells of interest is not only a good substitute for RNAi knockouts, which are known to be less efficient in insects and in many instances may cause a partial, hypomorphic phenotype (*Joga et al., 2016*), but also provides versatile possibilities as it can be used in combination with the powerful genetic tools of *Drosophila*.

Using this line, we could show that Dop1R2 is specifically required for later memory stages of both aversive and appetitive memory in the α/β-lobe and the α'/β'-lobe.

# Methods

## Key resources table

| Reagent type (species) or resource | Designation | Source or reference | Identifiers | Additional information |
|---|---|---|---|---|
| Genetic reagent *Drosophila melanogaster* both male and female | y[1] w[*] P{y[+t7.7]=nos-phiC31\int.NLS}X; P{y[+t7.7]=CaryIP}su(Hw)attP6 | Bloomington Stock Center | RRID:BDSC_32232 | |
| Genetic reagent *Drosophila melanogaster* both male and female | w;; Dr e/TM3 | Bloomington Stock Center | RRID:BDSC_36305 | |
| Genetic reagent *Drosophila melanogaster* both male and female | w[1118], 20XUAS-FLPG5.PEST | Bloomington Stock Center | RRID:BDSC_55807 | |
| Genetic reagent *Drosophila melanogaster* both male and female | OK107-Gal4 | Kyoto Stock Center | DGGR_106098 | |
| Genetic reagent *Drosophila melanogaster* both male and female | 5HTR1B-Gal4 | Bloomington Stock Center | RRID:BDSC_27636 | |
| Genetic reagent *Drosophila melanogaster* both male and female | c739-Gal4 | Hiromu Tanimoto (Tohoku University Japan) | RRID:BDSC_7362 | |
| Genetic reagent *Drosophila melanogaster* both male and female | c305a-Gal4 | Bloomington Stock Center | RRID:BDSC_30829 | |
| Genetic reagent *Drosophila melanogaster* both male and female | y w | Bloomington Stock Center | RRID:BDSC_1495 | |
| Genetic reagent *Drosophila melanogaster* both male and female | FRT-Dop1R2-HA-FRT Dop1R2$^{cko}$ | This paper | | See Materials and methods section: Creation of Dop1R2$^{cko}$ |
| Antibody | Mouse monoclonal α-HA clone 12CA5 | Roche | RRID:AB _514505 11583816001 | 1:200 |
| Antibody | Mouse monoclonal anti-Fasciclin II | Developmental Studies Hybridoma Bank | RRID:AB_528235 | 1:50 |
| Antibody | Rat monoclonal α-Droso-N-cadherin (Ncad) | Developmental Studies Hybridoma Bank | RRID:AB_528121 DN-Ex #8 | 1:30 |
| Recombinant protein | Recombinant DANN reagent, pCFD4-U6_1_6_3tandemgRNAs plasmid | Simon Bullock | RRID:Addgene_49411 | |

## Fly husbandry

*D. melanogaster* flies were reared in plastic vials on standard cornmeal food (12 g agar, 40 g sugar, 40 g yeast, 80 g cornmeal per liter) and transferred to fresh food vials every 2–3 days. Flies were generally kept at 25°C, 60–65% humidity, and exposed to 12 hr light and 12 hr darkness with light onset at 8 am. The following fly lines were used: *y[1] w[*] P{y[+t7.7]=nos-phiC31\int.NLS}X; P{y[+t7.7]=CaryIP} su(Hw)attP6* (abbreviated nos>Cas9 in this paper; BL 32232) for microinjection and as PCR template, *w;; Dr e/TM3* (BL 36305) was used as third chromosomal balancer line. *w[1118], 20XUAS-FLPG5.PEST* (BL 55807) was used as UAS-flp. OK107-Gal4 (106098) was obtained from the Kyoto Stock Center. The 5HTR1B-Gal4 line (BL 27636) and the c305a-Gal4 (BL 30829) line are from Bloomington Stock Center. The c739-Gal4 line was gifted to us by H Tanimoto (Tohoku University Japan). The *y w* (BL 1495) line was used as the control line.

**Table 1.** Primers and guide RNAs (gRNAs) for generating Dop1R2 conditional knockout lines.

| | Primer name | Sequence |
|---|---|---|
| Dop1R2 | | |
| Sequence-based reagent | Dop1R2_5'_fw_SpeI | GCTGCAGACTAGTCAGCCACCACA |
| | Dop1R2_5'_re_SmaI | CCTGAACCCGGGGATAAACTTTACCATAATGC |
| | Dop1R2_fragment_fw_AgeI | GTTCGACCGGTGATTGCATTGTGTTCACCAG |
| | Dop1R2_fragment_re_BstEII | GGGCTTGGTAACCACGACGAATCTTGCGTGGACAG |
| | Dop1R2_3'_fw_XhoI | ATTCTCGAGAAGTATCAACCCACGATGCGTTC |
| | Dop1R2_3'_re_Acc65I | GGCAACGGTACCAGATGCAGATACCG |
| Sequence-based reagent | Dop1R2_gRNA_5' | ACGAACTTAAGATAAAGTGTCGG |
| | Dop1R2_gRNA_3' | GCATCGTGGGCTGGTACTTCCGG |
| Sequence-based reagent | Dop1R2_Screening_for | TATCCCTATGACGTCCCGGAC |
| | Dop1R2_Screening_re | GACAGGTTGAGTGATGCGCC |
| | Dop1R2_5'FRT_for | GGCTACACATCATTTTATGCCAG |
| | Dop1R2_5'FRT_re | GTTCCTGTGCCTGATTCTGTTC |
| | Dop1R2_3'FRT_for | TCCTAACTGGCTTCTCTTCC |
| | Dop1R2_3'FRT_re | AGCGCTTAATTCACGAAAGC |

UAS-flp;; Dop1R2[cko] flies and Gal4;Dop1R2[cko] flies were crossed back with ;;Dop[cko] flies to obtain appropriate genetic controls, which were heterozygous for UAS and Gal4 but not Dop1R2[cko] to control for genetic background.

### Generation of Dop1R2[cko]

For generating the conditional knockout line, we needed three regions: (1) the 5' flanking sequence upstream of the first FRT site as homologous region, (2) the 3' flanking sequence downstream of the second FRT site as homologous region, (3) the sequence in between the two FRT sites hereafter named Dop1R2 coding fragment. To obtain the dopamine receptor fragments, genomic DNA from *nos>Cas9* flies was used as the template for the PCRs. The primers used for the different PCR fragments are shown in *Table 1*. The 3' and 5' fragments of Dop1R2 were subcloned into *pBluescript II SK(+)* vector (pBS) with adequate restriction enzymes – SpeI and SmaI for the 5' fragment and XhoI and Acc65I for the 3' fragment. The Dop1R2 coding fragment was cloned with the Invitrogen TOPO kit. After sequence confirmation, the fragments were cloned subsequently into pBS-FRT-3xHA-FRT. The vector is modified from the vector used by *Widmer et al., 2018*. The GFP was replaced by 3xHA using 3xHA Nco.

Fw: (cacatggtTACCCATACGATGTTCCTGACTATGCGGGCTATCCCTATGACGTCCCGGACTATGCAGGATCCTATCCATATGACGTTCCAGATTACGCTgca) and 3xHA H3

Re: (agcttgcAGCGTAATCTGGAACGTCATATGGATAGGATCCTGCATAGTCCGGGACGTCATAGGGATAGCCCGCATAGTCAGGAACATCGTATGGGTAc) as oligos, whereas the FRT site and the pBS backbone were kept. The Dop1R2 fragment was cloned in-frame in front of the 3xHA-tag using AgeI and BstEII-HF as restriction enzymes.

The two gRNAs were chosen using CRISPR Optimal Target Finder (http://targetfinder.flycrispr.neuro.brown.edu/). The used sequence was integrated into PCR primers to amplify a fragment of the *pCFD4-U6_1_6_3tandemgRNAs plasmid* (a gift from Simon Bullock, Addgene plasmid # 49411; *Port et al., 2014*) that was cloned into BbsI-digested pCFD4-U6_1_6_3tandemgRNAs vector using Gibson assembly (NEB). Insertion of the gRNAs was confirmed by sequencing. A mix containing 0.4 mg/ml template and 0.2 mg/ml of the gRNAs plasmid was injected into nos>Cas9 embryos. The injected flies were crossed with a third chromosomal balancer line. The F1 generation was crossed again with

the third chromosomal balancer flies. As soon as eggs or larvae were visible, the adults were sacrificed to check for the Dop1R2 construct via PCR. Positive hits were then sequenced, and a stock was established.

## Immunohistochemistry

Brains from 5- to 8-day-old flies were dissected in PBS (Bio-Froxx 1346LT050) and fixed in 3.7% formaldehyde for 25 min at RT. The brains were washed with 1X PBS containing 0.5% Triton X-100 (Carl Roth 3051.3) (0.5% PBST) before incubating the primary antibodies o/n at 4°C. The primary antibodies were mouse α-HA clone 12CA5 (Roche) 1:200, rat α-Droso-N-cadherin (Ncad) (Iowa H.B: DN-EX8) 1:30, or mouse α-Fasciclin II (FasII) (Iowa H.B: 1D4). After the brains were washed in 0.5% PBST again, they were incubated o/n at 4°C with the secondary antibodies. The following secondary antibodies were used: Goat α-rat Alexa 647 (Molecular Probes A-21247) and Goat α-Mouse Alexa 488 (Molecular Probes A11029) 1:200. The brains were washed again and mounted in self-made mounting media (90% Glycerol [Fischer Scientific Catalog No. BP229-1], 0.5% N-propyl gallate [Sigma P3130], 20 mM Tris [Fischer Scientific, Catalog No. BP152-5], pH 8.0) (adapted from NIC Harvard Medical School). The brains were imaged using a confocal microscope (Leica STELLARIS 8 FALCON) at ×40 magnification with the Plan APO ×40/1.10 water immersion objective at 1024×1024 pixels resolution and 600 Hz scan rate. Images were processed with Fiji ImageJ and Adobe Illustrator.

## Learning apparatus

For behavior experiments, we used a memory apparatus that is based on Tully and Quinn's design and modified to allow conducting four memory experiments in parallel (CON-Elektronik, Greussenheim, Germany). Experiments were performed at 23–25°C and 65–75% relative humidity. The training was performed in dim red light, and memory tests were done in complete darkness. The two odors used were 3-octanol (3-Oct) (Sigma-Aldrich 218405) and 4-methyl-cyclohexanol (MCH) (Sigma-Aldrich 66360) diluted in paraffin oil (Sigma-Aldrich 18512) 1:100, respectively. 260 µl of the diluted odors were presented in a plastic cup of 14 mm in diameter. A vacuum membrane pump ensured odor delivery at a flow rate of 7 l/min.

## Aversive olfactory conditioning

For aversive conditioning, groups of 50–100 flies of mixed sex were loaded in tubes lined with an electrifiable copper grid. The position in the machine and the sequence in which the genotypes were tested were randomized. Experiments in which more than half of the flies died, the flies did not move, or there were technical problems with the machine, as well as human errors, were excluded. The training was conducted in the morning. After an accommodation period of 90 s, the first odor was presented for 60 s. In parallel, 12 pulses of 100 V for 1.5 s were delivered with an interval of 3.5 s. After 30 s of flushing with fresh air, the second odor was presented for 60 s. For the subsequent group of flies, the order of the two odors was reversed. For measuring 0 hr performance, flies were tested about 3 min after the end of the conditioning. To determine 2 hr memory performance, flies were transferred to food vials after conditioning and kept at 25°C until the test. For each genotype and condition, the biological replicate is N=12.

For long-term aversive memory, groups of flies were trained with six cycles, where each cycle was composed of 60 s of the first odor presented simultaneously with 12 pulses of 90 V for 1.5 s with an interval of 3.5 s followed by 30 s of fresh air and 60 s of the second odor. The cycles were spaced by 15 min of inter-training intervals. The flies were starved for 24 hr before testing for LTM to increase motivation to climb the Y-Maze and make a decision. The biological replicate is N=7.

## Appetitive olfactory conditioning

Before appetitive conditioning, groups of 50–100 flies with mixed sex were starved for 19–21 hr in plastic vials containing damp cotton at the bottom. Experiments in which more than half of the flies died, the flies did not move, or there were technical problems with the machine, as well as human errors, were excluded. The position in the machine and the sequence in which the genotypes were tested were randomized. The training was conducted in the morning. The conditioning protocol consists of a 90 s accommodation period, 120 s of the first odor, 60 s of fresh air followed by 120 s of the second odor. During the first odor, flies are in a conditioning tube lined with filter

paper that was soaked in water the day before the experiment and left to dry overnight. For the second odor, flies are transferred to a conditioning tube lined with a filter paper that was soaked with a 1.5 M sucrose (Sigma-Aldrich, Cat# 84100-1KG; CAS Number 57-50-1) solution on the day before and left to dry at RT. After conditioning, flies were either directly tested for STM or put back in starvation vials until the memory test 2 hr later. For 24 hr memory, flies were fed for 3 hr after training before starving them again. One experiment consisted of two reciprocal conditionings, in which the odor paired with sucrose was reversed. For each genotype and condition, the biological replicate is N=12.

### Memory tests

Flies were loaded into a sliding compartment and transferred to a two-arm choice point. Animals were allowed to choose between 3-octanol and 4-methyl-cyclohexanol. After 60 s, flies trapped in both arms were collected separately and counted. Based on these numbers, a preference index was calculated as follows: PREF = $((N_{arm1} - N_{arm2})\,100)/N_{total}$. The two preference indices were calculated from the two reciprocal experiments. The average of these two PREFs gives a learning index (LI). LI = $(PREF_1 + PREF_2)/2$.

In case of all long-term aversive memory experiments, the Y-Maze protocol was adapted to test flies 24 hr post-training. Testing using the Y-Maze was done following the protocol as described in *Mohandasan et al., 2022*, where flies were loaded at the bottom of 20 min odorized 3D-printed Y-Mazes from where they would climb up to a choice point and choose between the two odors. The learning index was then calculated after counting the flies in each odorized vial as follows: LI = $((N_{CS-} - N_{CS+})\,100)/N_{total}$, where $N_{CS-}$ and $N_{CS+}$ are the number of flies that were found trapped in the untrained and trained odor tube, respectively.

### Sensory accuracy tests

Flies were tested for their ability to sense the two used odors 3-octanol and 4-methyl-cyclohexanol, as well as electric shock. Therefore, the flies were loaded into a sliding compartment and brought to a two-arm choice point. The flies were allowed to freely choose between an arm containing the stimulus and a neutral arm. All experiments were carried out in the dark. Afterward, the flies in each arm were counted, and a preference index was calculated.

For testing the odor response, the flies could choose between one of the odors in the same concentration as used for the behavior experiment and the same amount of paraffin oil for 120 s.

Preference index PI = $((N_{air} - N_{odor})100)/N_{total}$.

For shock response, the flies could freely choose for 60 s between a copper-grid lined tube getting pulses of 100 V or a copper-grid lined tube getting no electric shock. Preference index PI = $((N_{No\ shock} - N_{shock})100)/N_{total}$.

For testing sugar sensitivity, a group of flies was starved for 1–21 hr in a tube with damp cotton on the bottom. They could choose for 120 s between a tube lined with filter paper that was soaked in 1.5 M sucrose solution the day before or a tube lined with filter paper that was soaked in distilled water the day before. Preference index PI = $((N_{sucrose} - N_{water})100)/N_{total}$.

### Statistical analysis

To compare performance indices between different groups, we used one-way ANOVA (analysis of variance) with post hoc Tukey's HSD (honestly significant difference) test calculator for comparing multiple treatments in R with the package multcomp. In the case of two groups, we performed a t-test for comparison.

To verify if the data was normally distributed, we used Shapiro-Wilk's test for normal distribution on the statistics software, Prism GraphPad.

We used one-way ANOVA with post hoc Tukey's HSD test when the data was normally distributed. And the nonparametric counterpart, Kruskal-Wallis test when it wasn't, along with Dunn's multiple comparisons test. We also verified if all the groups were significantly different from zero using a one-sample t-test for normally distributed data and a Wilcoxon's signed-rank test when the data was not normally distributed.

## Single-cell transcriptomic data analysis

Single-cell transcriptomics data from *Davie et al., 2018*, was downloaded from the Gene Expression Omnibus (GEO). The metadata containing the annotations was also downloaded and used to annotate the clusters. The clustering and further transcriptomic analyses were performed using the Seurat (version 5.2.1) package for R (*Hao et al., 2024*). The clustering was done at 0.5 Seurat resolution. And the UMAP was computed using the top 10 principal components. The LogNormalize method from Seurat was used to calculate log-normalized gene expression values. The Venn diagram in *Figure 1—figure supplement 1* was created using BioVenn (*Hulsen et al., 2008*).

## Acknowledgements

We would like to thank H Tanimoto, the Kyoto and Bloomington stock centers for fly strains. We would like to thank Dr. Christine Guzman for help with transcriptomics analyses, Christopher Aeschbacher and the Bioimage Core Facility for help with imaging. We would like to thank colleagues of the Sprecher lab for their valuable input and discussions.

## Additional information

### Funding

| Funder | Grant reference number | Author |
| --- | --- | --- |
| Novartis Stiftung für Medizinisch-Biologische Forschung | #23B138 | Simon G Sprecher |
| Schweizerischer Nationalfonds zur Förderung der Wissenschaftlichen Forschung | 310030_219348 | Simon G Sprecher |
| Swiss National Science Foundation | IZKSZ3_218514 | Simon G Sprecher |

The funders had no role in study design, data collection and interpretation, or the decision to submit the work for publication.

### Author contributions

Jenifer C Kaldun, Conceptualization, Formal analysis, Validation, Investigation, Visualization, Writing – original draft; Emanuele Calia, Ganesh Chinmai Bangalore Mukunda, Validation, Investigation, Visualization, Writing – review and editing; Cornelia Fritsch, Investigation, Methodology, Writing – original draft, Writing – review and editing; Nikita Komarov, Investigation, Writing – original draft, Writing – review and editing; Simon G Sprecher, Conceptualization, Supervision, Funding acquisition, Writing – original draft, Project administration, Writing – review and editing

### Author ORCIDs

Jenifer C Kaldun ⓘD https://orcid.org/0000-0003-3385-366X
Emanuele Calia ⓘD https://orcid.org/0009-0007-4121-9526
Ganesh Chinmai Bangalore Mukunda ⓘD https://orcid.org/0009-0001-3850-9801
Nikita Komarov ⓘD https://orcid.org/0000-0002-6592-9238
Simon G Sprecher ⓘD https://orcid.org/0000-0001-9060-3750

Reviewer #1 (Public review): https://doi.org/10.7554/eLife.99368.3.sa1
Reviewer #2 (Public review): https://doi.org/10.7554/eLife.99368.3.sa2
Reviewer #3 (Public review): https://doi.org/10.7554/eLife.99368.3.sa3
Author response https://doi.org/10.7554/eLife.99368.3.sa4

# Additional files

## Supplementary files
MDAR checklist

## Data availability
All data generated or analysed during this study are included in the manuscript and supporting files. All used data has been deposited in a Zenodo repository (*Kaldun et al., 2025*).

The following dataset was generated:

| Author(s) | Year | Dataset title | Dataset URL | Database and Identifier |
|---|---|---|---|---|
| Kaldun JC, Bangalore Mukunda GC, Calia E, Fritsch C, Komarov N, Sprecher S | 2025 | Data for: A temporally restricted function of the Dopamine receptor Dop1R2 during memory formation | https://doi.org/10.5281/zenodo.15772120 | Zenodo, 10.5281/zenodo.15772120 |

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
