## [Editor Report · eLife Assessment]

The authors design and implement an elegant strategy to delete genomic sequences encoding the dopamine receptor dop1R2 from specific subsets of mushroom body neurons (ab, a'b' and gamma) and show that while none of these manipulations affect short term appetitive or aversive memory, loss of dop1R2 from ab or a'b' block the ability of flies to display measurable forms of longer forms of memory. These findings are **important** in confirming and extending prior observations, and well supported by **convincing** evidence that build on precise techniques for genetic perturbation.

---

## [Referee Report · Reviewer #1 (Public review)]

Summary:

In this manuscript the authors present a novel CRISPR/Cas9-based genetic tool for the dopamine receptor dop1R2. Based on the known function of the receptor in learning and memory, they tested the efficacy of the genetic tool by knocking out the receptor specifically in mushroom body neurons. The data suggest that dop1R2 is necessary for longer lasting memories through its action on ⍺/ß and ⍺'/ß' neurons but is dispensable for short-term memory and thus in ɣ neurons. The experiments impressively demonstrate the value of such a genetic tool and illustrate the specific function of the receptor in subpopulations of KCs for longer-term memories.

---

## [Referee Report · Reviewer #2 (Public review)]

Summary:

This manuscript examines the role of the dopamine receptor, Dop1R2, in memory formation. This receptor has complex roles in supporting different stages of memory, and the neural mechanisms for these functions is poorly understood. The authors are able to localize Dop1R2 function to the vertical lobes of the mushroom body, revealing a role in later (presumably middle-term) aversive and appetitive memory. In general the experimental design is rigorous, and statistics are appropriately applied. The manuscript provides a thorough assessment of how Dop1R2 functions within the mushroom bodies to regulate protein-synthesis dependent and independent memory, and provides a valuable new tool for the community.

Strengths:

(1) The FRT lines generated provide a novel tool for temporal and spatially precise manipulation of Dop1R2 function. This tool will be valuable to study the role of Dop1R2 in memory and other behaviors potentially regulated by this gene.

(2) Given the highly conserved role of Dop1R2 in memory and other processes, these findings have high potential to translate to vertebrate species.

---

## [Referee Report · Reviewer #3 (Public review)]

Summary:

Kaldun et al. investigated the role of Dopamine Receptor Dop1R2 in different types and stages of olfactory associative memory in *Drosophila melanogaster*. Dop1R2 is a type 1 Dopamine receptor that can act both through Gs-cAMP and Gq-ERCa2+ pathways. The authors first developed a sophisticated tool where tissue-specific knock-out mutants can be generated using Crispr/Cas9 technology in combination with the Gal4/UAS gene-expression toolkit.They direct the K.O. mutation to intrinsic neurons of the main associative memory centre fly brain: the mushroom body (MB). There are three main types of MB-neurons, or Kenyon cells, according to their axonal projections: a/b; a'/b' and g neurons.

Kaldun et al. found that, while not required for short-term memory, dop1R2 is necessary in a/b and a'/b' but not in gamma neurons to display normal appetitive and aversive middle-term (2h) and long-term (24h) memory. These results showcase a compartmentalized role of Dop1R2 in specific neuronal subtypes of the main memory centre of the fly brain for the expression of middle and long-term memories.

The conclusions of this paper are very well supported by the data, and the authors systematically addressed the requirement of a very interesting type of dopamine receptor in both appetitive and aversive memories. These findings are important for the fields of learning and memory and dopaminergic neuromodulation, among others.

Importantly, the authors of this paper produced a tool to generate tissue-specific knock out mutants of dop1R2. Although reports on the requirement of this gene in different memory phases exist, the genetic tools used here represent the most sophisticated approach to induce a loss of function phenotypes in neurons of interest.

Overall, the authors generated a very useful tool to study dopamine neuromodulation in any given circuit when used in combination with the powerful genetic toolkit available in *Drosophila*. The reports on this paper confirmed a previously described role of Dop1R2 in the expression of aversive and appetitive LTM providing spatio-temporal resolution and additionally, they mapped these effects to two types of memory neurons in the fly brain, shedding light into the intricate modulation of dopamine in memory circuits.

---

## [Author Response]

The following is the authors’ response to the original reviews.

**Public Reviews:**

**Reviewer #1 (Public Review):**
Summary:In this manuscript, the authors present a novel CRISPR/Cas9-based genetic tool for the dopamine receptor dop1R2. Based on the known function of the receptor in learning and memory, they tested the efficacy of the genetic tool by knocking out the receptor specifically in mushroom body neurons. The data suggest that dop1R2 is necessary for longer-lasting memories through its action on ⍺/ß and ⍺'/ß' neurons but is dispensable for short-term memory and thus in ɣ neurons. The experiments impressively demonstrate the value of such a genetic tool and illustrate the specific function of the receptor in subpopulations of KCs for longer-term memories. The data presented in this manuscript are significant.
**Reviewer #2 (Public Review):**
Summary:This manuscript examines the role of the dopamine receptor, Dop1R2, in memory formation. This receptor has complex roles in supporting different stages of memory, and the neural mechanisms for these functions are poorly understood. The authors are able to localize Dop1R2 function to the vertical lobes of the mushroom body, revealing a role in later (presumably middle-term) aversive and appetitive memory. In general, the experimental design is rigorous, and statistics are appropriately applied. While the manuscript provides a useful tool, it would be strengthened further by additional mechanistic studies that build on the rich literature examining the roles of dopamine signaling in memory formation. The claim that Dop1R2 is involved in memory formation is strongly supported by the data presented, and this manuscript adds to a growing literature revealing that dopamine is a critical regulator of olfactory memory. However, the manuscript does not necessarily extend much beyond our understanding of Dop1R2 in memory formation, and future work will be needed to fully characterize this reagent and define the role of Dop1R2 in memory.Strengths:(1) The FRT lines generated provide a novel tool for temporal and spatially precise manipulation of Dop1R2 function. This tool will be valuable to study the role of Dop1R2 in memory and other behaviors potentially regulated by this gene.(2) Given the highly conserved role of Dop1R2 in memory and other processes, these findings have a high potential to translate to vertebrate species.Weaknesses:(1) The authors state Dop1R2 associates with two different G-proteins. It would be useful to know which one is mediating the loss of aversive and appetitive memory in Dop1R2 knockout flies.

We thank you for the insightful comment. We agree that it would be very useful to know which G-proteins are transmitting Dop1R2 signaling. To that extent, we examined single-cell transcriptomics data to check the level of co-expression of Dop1R2 with G-proteins that are of interest to us. (Figure 1 S1)

Lines 312-325

“Some RNA binding proteins and Immediate early genes help maintain identities of Mushroom body cells and are regulators of local transcription and translation (de Queiroz et al., 2025; Raun et al., 2025). So, the availability of different G-proteins may change in different lobes and during different phases of memory. The G-protein via which GPCRs signal, may depend on the pool of available G-proteins in the cell/sub-cellular region (Hermans, 2003)., Therefore, Dop1R2 may signal via different G-proteins in different compartments of the Mushroom body and also different compartments of the neuron. We looked at Gα_o_ and Gα_q_ as they are known to have roles in learning and forgetting (Ferris et al., 2006; Himmelreich et al., 2017). We found that Dop1R2 co-expresses more frequently with Gα_o_ than with Gα_q_ (Figure 1 S1). While there is evidence for Dop1R2 to act via Gα_q_ (Himmelreich et al., 2017). It is difficult to determine whether this interaction is exclusive, or if Dop1R2 can also be coupled to other G-proteins. It will be interesting to determine the breadth of G-proteins that are involved in Dop1R2 signaling.”

(2) It would be interesting to examine 24hr aversive memory, in addition to 24hr appetitive memory.

This is indeed an important point and we agree that it will complete the assessment of temporally distinct memory traces. We therefore performed the Aversive LTM experiments and include them in the results.

Lines 208-228

“24h memory is impaired by loss of Dop1R2

Next, we wanted to see if later memory forms are also affected. One cycle of reward training is sufficient to create LTM (Krashes & Waddell, 2008), while for aversive memory, 5-6 cycles of electroshock-trainings are required to obtain robust long-term memory scores (Tully et al., 1994). So, we looked at both, 24h aversive and appetitive memory. For aversive LTM, the flies were tested on the Y-Maze apparatus as described in (Mohandasan et al., 2022).

Flipping out Dop1R2 in the whole MB causes a reduced 24h memory performance (Figure 4A, E). No phenotype was observed when Ddop1R2 was flipped out in the γ-lobe (Figure 4B, F). However, similar to 2h memory, loss of Ddop1R2 in the α/β-lobes (Figure 4C, G) or the α’/β’-lobes (Figure 4D, H) causes a reduction in memory performance. Thus, Dop1R2 seems to be involved in aversive and appetitive LTM in the α/β-lobes and the α’/β’-lobes.

Previous studies have shown mutation in the Dop1R2 receptor leads to improvement in LTM when a single shock training paradigm is used (Berry et al., 2012). As we found that it disrupts LTM, we wanted to verify if the absence of Dop1R2 outside the MB is what leads to an improvement in memory. To that extent, we tested panneuronal flip-out of Dop1R2 flies for 6hr and 24hr memory upon single shock using the elav-Gal4 driver. We found that it did not improve memory at both time points (Figure 4 S1). Confirming that flipping out Dop1R2 panneuronally does not improve LTM (Figure 4 S1C) and highlighting its irrelevance in memory outside the MB.”

(3) The manuscript would be strengthened by added functional analysis. What are the DANs that signal through Dop1R. How do these knockouts impact MBONs?

We thank you for this question. We indeed agree that it is a highly relevand and open question, how distinct DANs signal via distinct Dopamine receptors. Our work here uniquely focusses on Dop1R2 within the MB. We aim to investigate other DopRs and the connection between DANs in the future using similar approaches.

(4) Also in Figure 2, the lobe-specific knockouts might be moved to supplemental since there is no effect. Instead, consider moving the control sensory tests into the main figure.

We thank you for this suggestion and understand that in Figure 2 no significant difference is seen. However, we have emphasized in the text that the results from the supplementary figures are just to confirm that the modifications made at the Dop1R2 locus did not alter its normal function.

Lines 156-162

“We wanted to see if flipping out Dop1R2 in the MB affects memory acquisition and STM by using classical olfactory conditioning. In short, a group of flies is presented with an odor coupled to an electric shock (aversive) or sugar (appetitive) followed by a second odor without stimulus. For assessing their memory, flies can freely choose between the odors either directly after training (STM) or at a later timepoint.

To ensure that the introduced genetic changes to the Dop1R2 locus do not interfere with behavior we first checked the sensory responses of that line”

(5) Can the single-cell atlas data be used to narrow down the cell types in the vertical lobes that express Dop1R2? Is it all or just a subset?

This is indeed an interesting question, and we thank you for mentioning it. To address this as best as we could, we analyzed the single cell transcriptomic data from (Davie et al., 2018) and presented it in Figure 1 S1.

**Reviewer #3 (Public Review):**
Summary:Kaldun et al. investigated the role of Dopamine Receptor Dop1R2 in different types and stages of olfactory associative memory in *Drosophila melanogaster*. Dop1R2 is a type 1 Dopamine receptor that can act both through Gs-cAMP and Gq-ERCa2+ pathways. The authors first developed a very useful tool, where tissue-specific knock-out mutants can be generated, using Crispr/Cas9 technology in combination with the powerful Gal4/UAS gene-expression toolkit, very common in fruit flies.They direct the K.O. mutation to intrinsic neurons of the main associative memory centre fly brain-the mushroom body (MB). There are three main types of MB-neurons, or Kenyon cells, according to their axonal projections: a/b; a'/b', and g neurons.Kaldun et al. found that flies lacking dop1R2 all over the MB displayed impaired appetitive middle-term (2h) and long-term (24h) memory, whereas appetitive short-term memory remained intact. Knocking-out dop1R2 in the three MB neuron subtypes also impaired middle-term, but not short-term, aversive memory.These memory defects were recapitulated when the loss of the dop1R2 gene was restricted to either a/b or a'/b', but not when the loss of the gene was restricted to g neurons, showcasing a compartmentalized role of Dop1R2 in specific neuronal subtypes of the main memory centre of the fly brain for the expression of middle and long-term memories.Strengths:(1) The conclusions of this paper are very well supported by the data, and the authors systematically addressed the requirement of a very interesting type of dopamine receptor in both appetitive and aversive memories. These findings are important for the fields of learning and memory and dopaminergic neuromodulation among others. The evidence in the literature so far was generated in different labs, each using different tools (mutants, RNAi knockdowns driven in different developmental stages...), different time points (short, middle, and long-term memory), different types of memories (Anesthesia resistant, which is a type of protein synthesis independent consolidated memory; anesthesia sensitive, which is a type of protein synthesis-dependent consolidated memory; aversive memory; appetitive memory...) and different behavioral paradigms. A study like this one allows for direct comparison of the results, and generalized observations.(2) Additionally, Kaldun and collaborators addressed the requirement of different types of Kenyon cells, that have been classically involved in different memory stages: g KCs for memory acquisition and a/b or a'/b' for later memory phases. This systematical approach has not been performed before.(3) Importantly, the authors of this paper produced a tool to generate tissue-specific knock-out mutants of dop1R2. Although this is not the first time that the requirement of this gene in different memory phases has been studied, the tools used here represent the most sophisticated genetic approach to induce a loss of function phenotypes exclusively in MB neurons.Weaknesses:(1) Although the paper does have important strengths, the main weakness of this work is that the advancement in the field could be considered incremental: the main findings of the manuscript had been reported before by several groups, using tissue-specific conditional knockdowns through interference RNAi. The requirement of Dop1R2 in MB for middle-term and long-term memories has been shown both for appetitive (Musso et al 2015, Sun et al 2020) and aversive associations (Plaçais et al 2017).

Thank you for this comment. We believe that the main takeaway from the paper is the elegant tool we developed, to study the role of Dop1R2 in fruit flies by effectively flipping it out spatio-temporally. Additionally, we studied its role in all types of olfactory associative memory to establish it as a robust tool that can be used for further research in place of RNAi knockouts which are shown to be less efficient in insects as mentioned in the texts in line 394-398.

“The genetic tool we generated here to study the role of the Dop1R2 dopamine receptor in cells of interest, is not only a good substitute for RNAi knockouts, which are known to be less efficient in insects (Joga et al., 2016), but also provides versatile possibilities as it can be used in combination with the powerful genetic tools of *Drosophila*.”

(2) The approach used here to genetically modify memory neurons is not temporally restricted. Considering the role of dopamine in the correct development of the nervous system, one must consider the possible effects that this manipulation can have in the establishment of memory circuits. However, previous studies addressing this question restricted the manipulation of Dop1R2 expression to adulthood, leading to the same findings than the ones reported in this paper for both aversive and appetitive memories, which solidifies the findings of this paper.

We thank you for this comment and we agree that it would be important to show a temporally restricted effect of Dop1R2 knockout. To assess this and rule out potential developmental defects we decided to restrict the knockout to the post-eclosion stage and to include these results.

Lines 230-250

“Developmental defects are ruled out in a temporally restricted Dop1R2 conditional knockout.

To exclude developmental defects in the MB caused by flip-out of Dop1R2, we stained fly brains with a FasII antibody. Compared to genetic controls, flies lacking Dop1R2 in the mushroom body had unaltered lobes (Figure 4 S2C).

Regardless, we wanted to control for developmental defects leading to memory loss in flip-out flies. So, we generated a Gal80ts-containing line, enabling the temporal control of Dop1R2 knockout in the entire mushroom body (MB). Given that the half-life of the receptor remains unknown, we assessed both aversive short-term memory (STM) and long-term memory (LTM) to determine whether post-eclosion ablation of Dop1R2 in the MB produced differences compared to our previously tested line, in which Dop1R2 was constitutively knocked out from fertilization. To achieve this, flies were maintained at 18°C until eclosion and subsequently shifted to 30°C for five to seven days. On the fifth day, training was conducted, followed by memory testing. Our results indicate that aversive STM was not significantly impaired in Dop1R2-deficient MBs compared to control flies (Figure 4 S3), consistent with our previous findings (Figure 2). However, aversive LTM was significantly impaired relative to control lines (Figure 4 S3), which also aligned with prior observations. These findings strongly indicate that memory loss caused by Dop1R2 flip-out is not due to developmental defects.”

(3) The authors state that they aim to resolve disparities of findings in the field regarding the specific role of Dop1R2 in memory, offering a potent tool to generate mutants and addressing systematically their effects on different types of memory. Their results support the role of this receptor in the expression of long-term memories, however in the experiments performed here do not address temporal resolution of the genetic manipulations that could bring light into the mechanisms of action of Dop1R2 in memory. Several hypotheses have been proposed, from stabilization of memory, effects on forgetting, or integration of sequences of events (sensory experiences and dopamine release).

We thank you for this comment. We agree that it would be interesting to dissect the memory stages by knocking out the receptor selectively in some of them (encoding, consolidation, retrieval). However, our tool irreversibly flips out Dop1R2 preventing us from investigating the receptor’s role in retrieval. Our results show that the receptor is dispensable for STM formation (Figure 2, Figure 4 Supplement 3), suggesting that it is not involved in encoding new information. On the other hand, it is instead involved in consolidation and/or retrieval of long-term and middle-term memories (Figure 3, Figure 4, Figure 5B).

Overall, the authors generated a very useful tool to study dopamine neuromodulation in any given circuit when used in combination with the powerful genetic toolkit available in *Drosophila*. The reports in this paper confirmed a previously described role of Dop1R2 in the expression of aversive and appetitive LTM and mapped these effects to two specific types of memory neurons in the fly brain, previously implicated in the expression and consolidation of long-term associative memories.
**Recommendations for the authors:**

**Reviewer #1 (Recommendations For The Authors):**
(1) On the first view, the results shown here are different from studies published earlier, while in the same line with others (e.g. Sun et al, for appetitive 24h memories). For example, Berry et al showed that the loss of dop1R2 impairs immediate memory, while memory scores are enhanced 3h, 6h, and 24h after training. Further, they showed data that shock avoidance, at least for higher shock intensities, is reduced in mutant (damb) flies. All in all, this favors how important it is to improve the genetic tools for tissue-specific manipulation. Despite the authors nicely discussing their data with respect to the previous studies, I wondered whether it would be suitable to use the new tool and knock out dop1R2 panneuronally to see whether the obtained data match the results published by Berry et al.. Further, as stated in line 105ff: "As these studies used different learning assays - aversive and appetitive respectively as well as different methods, it is unclear if Dop1R2 has different functions for the different reinforcement stimulus" I wondered why the authors tested aversive and appetitive learning for STM and 2h memory, but only appetitive memory for 24h.

Thank you for this comment. To that extent, as mentioned above in response to reviewer #2, we included in the results the aversive LTM experiment (Figure 4). Moreover, we performed experiments along the line of Berry et al. using our tool as shown in Figure 4 S1. Our results support that Dop1R2 is required for LTM, rather than to promote forgetting.

(2) Line 165ff: I can´t find any of the supplementary data mentioned here. Please add the corresponding figures.

Thank you for pointing this out. In that line we don’t refer to any supplementary data, but to the Figure 1F, showing the absence of the HA-tag in our MB knock-out line. We have clarified this in the text (lines 151-153)

(3) I can't imagine that the scale bar in Figure 1D-F is correct. I would also like to suggest to show a more detailed analysis of the expression pattern. For example, both anterior and posterior views would be appropriate, perhaps including the VNC. This would allow the expression pattern obtained with this novel tool to be better compared with previously published results. Also, in relation to my comment above (1), it may help to understand the functional differences with previous studies, especially as the authors themselves state that the receptor is "mainly" expressed in the mushroom body (line 99). It would be interesting to see where else it is expressed (if so). This would also be interesting for the panneuronal knockdown experiment suggested under (1). If the receptor is indeed expressed outside the mushroom body, this may explain the differences to Berry et al.

Thank you for noting this, there was indeed a mistake in the scale bar which we now fixed. Since with our HA-tag immunostaining we could not detect any noticeable signal outside of the MB, we decided to analyze previously existing single cell transcriptomics data that showed expression of the receptor in 7.99% of cells in the VNC and in 13.8% of cells outside the MB (lines 98-100) confirming its sparse expression in the nervous system. The lack of detection of these cells is likely due to the sparse and low expression of the protein. The HA-tag allows to detect the endogenous level of the locus (it is possible that a Gal4/UAS amplification of the signal might allow to detect these cells).

Regarding the panneuronal knockout, we decided to try to replicate the experiment shown in Berry et al. in Figure 4 S1 and found that Dop1R2 is required for LTM.

(4) Related to learning data shown in Figures 2-4, the authors should show statistical differences between all groups obtained in the ANOVA + PostHoc tests. Currently, only an asterisk is placed above the experimental group, which does not adequately reflect the statistical differences between the groups. In addition, I would like to suggest adding statistical tests to the chance level as it may be interesting to know whether, for example, scores of knockout flies in 3C and 3D are different from the chance level.

Many thanks for this correction, we agree with the fact that the way significance scores were shown was not informative enough. We fixed the point by now showing significance between all the control groups and the experimental ones. We also inserted the chance level results in the figure legends.

(5) Unfortunately, the manuscript has some typing errors, so I would like to ask the authors to check the manuscript again carefully.Some Examples:Line 31: the theLine 56: G-ProteinLine 64: c-AMPLine 68: DopamineLine 70: G-Protein (It alternates between G-protein and G-Protein)Line 76: References are formatted incorrectlyLine 126: Ha-Tag (It alternates between Ha and HA)Line 248: missing space before the bracket...is often found

Thank you for noticing these errors, we have now corrected the spelling throughout the manuscript.

(6) In the figures the axes are labelled Preference Index (Pref"I"). In the methods, however, the calculation formula is defined as "PREF".

We thank you for drawing attention to this. To avoid confusion, we changed the definition in the methods section so that it could be clear and coherent (“Memory tests” paragraph in the methods section).

“PREF = ((N_arm1_ - N_arm2_) 100) / N_total_ the two preference indices were calculated from the two reciprocal experiments. The average of these two PREFs gives a learning index (LI). LI = (PREF_1_ + PREF_2_) / 2.

In case of all Long-term Aversive memory experiments, Y-Maze protocol was adapted to test flies 24 hours post training. Testing using the Y-Maze was done following the protocol as described in (Mohandasan et al., 2022) where flies were loaded at the bottom of 20-minutes odorized 3D-printed Y-Mazes from where they would climb up to a choice point and choose between the two odors. The learning index was then calculated after counting the flies in each odorized vial as follows: LI = ((N_CS-_ - N_CS+_) 100) / N_total_. Where NCS- and NCS+ are the number of flies that were found trapped in the untrained and trained odor tube respectively.

**Reviewer #2 (Recommendations For The Authors):**
(1) In Figures 2 and 3, the legends running two different subfigures is confusing. Would be helpful to find a different way to present.

Thank you for your suggestion. We modified how we present legends, placing them vertically so that it is clearer.

(2) Use additional drivers to verify middle and long-term memory phenotypes.

We agree that it would be interesting to see the role of Dop1R2 in other neurons. To that extent, we looked at long term aversive memory in flies where the receptor was panneuronaly flipped out, and did not find evidence that suggested involvement of Dop1R2 in memory processes outside the MB. (Figure 4 S1)

(3) Additional discussion of genetic background for fly lines would be helpful.

Thank you for your advice. We have mentioned the genetic background of flies in the key resources table of the methods sections. Additionally, we also included further explanation on how the lines were created and their genetic background (see “Fly Husbandry” paragraph in the methods section).

“UAS-flp;;Dop1R2 cko flies and Gal4;Dop1R2^cko^ flies were crossed back with ;;Dop^cko^ flies to obtain appropriate genetic controls which were heterozygous for UAS and Gal4 but not Dop1R2^cko^.”

**Reviewer #3 (Recommendations For The Authors):**
Line 109 states that to resolve the problem a tool is developed to knock down Dop1R2 in s spatial and temporal specific manner- while I agree that this is within the potential of the tool, there is no temporal control of the flipase action in this study; at least I cannot find references to the use of target/gene switch to control stages of development or different memory phases. However the version available for download is missing supplementary information, so I did not have access to supplementary figures and tables.

Thank you for the comment, as mentioned before it would be great to be able to dissect the memory phases. We show in lines 232 – 250 and Figure 4 S3 that the temporally restricted flip-out to the post-eclosion life stage gave us coherent results with the previous findings, ruling out potential developmental defects.

In relation to my comment on the possible developmental effects of the loss of the gene, Figure 1F could showcase an underdeveloped g lobe when looking at the lobe profiles. I understand this is not within the scope of the figure, but maybe a different z projection can be provided to confirm there are no obvious anatomical alterations due to the loss of the receptor.

We understand the doubt about the correct development of the MB and we thank you for your insightful comment. To that extent we decided to perform a FasII immunostaining that could show us the MB in the different lines (Figure 4 S2) and it appears that there are no notable differences in the lobes development in our knockout line.

It seems that the obvious missing piece of the puzzle would be to address the effects of knocking out Dop1R2 in aversive LTM. The idea of systematically addressing different types of memory at different time points and in different KCs is the most attractive aspect of this study beyond the technical sophistication, and it feels that the aim of the study is not delivered without that component.

We agree and we thank you for the clarification. As mentioned above in response to Reviewer #2, we decided to test aversive LTM as described in lines –208-228, Figure 4, Figure 4 S1.

Some statements of the discussion seem too vague, and I think could benefit from editing:Line 284 "however other receptors could use Gq and mediate forgetting"- does this refer to other dopamine receptors? Other neuromodulators? Examples?

Thank you for pointing this out. We Agree and therefore decided to omit this line.

Line 289 "using a space training protocol and a Dop1R2 line" - this refers to RNAi lines, but it should be stated clearly.

That is correct, we thank you for bringing attention to this and clarified it in the manuscript.

–Lines 329-330

“Interestingly, using a spaced training protocol and a Dop1R2 RNAi knockout line another study showed impaired LTM (Placais et al., 2017).”

The paragraph starting in line 305 could be re-written to improve clarity and flow. Some statements seem disconnected and require specific citations. For example "In aversive memory formation, loss of Dop1R2 could lead to enhanced or impaired memory, depending on the activated signaling pathways and the internal state of the animal...". This is not accurate. Berry et al 2012 report enhanced LTM performance in dop1R2 mutants whereas Plaçais et al 2017 report LTM defects in Dop1R2 knock-downs, but these different findings do not seem to rely on different internal states or signaling pathways. Maybe further elaboration can help the reader understand this speculation.

We agree and we thank you for this advice. We decided to add additional details and citations to validate our speculation

Lines 350-353

“In aversive memory formation, loss of Dop1R2 could lead to enhanced or impaired memory, depending on the activated signaling pathways. The signaling pathway that is activated further depends on the available pool of secondary messengers in the cell (Hermans, 2003) which may be regulated by the internal state of the animal.”

"...for reward memory formation, loss of Dop1R2 seems to impair memory", this seems redundant at this point, as it has been discussed in detail, however, citations should be provided in any case (Musso 2015, Sun 2020)

Thank you for noting this. We recognize the redundancy and decided to exclude the line.

Finally, it would be useful to additionally refer to the anatomical terminology when introducing neuron names; for example MBON MVP2 (MBON-g1pedc>a/b), etc.

Thank you for this suggestion. We understand the importance of anatomical terminologies for the neurons. Therefore, we included them when we introduce neurons in the paper.

We thank you for your observations. We recognize their value, so we have made appropriate changes in the discussion to sound less vague and more comprehensive.